# Chromatin Organization and Function in *Drosophila*

**DOI:** 10.3390/cells10092362

**Published:** 2021-09-08

**Authors:** Palmira Llorens-Giralt, Carlos Camilleri-Robles, Montserrat Corominas, Paula Climent-Cantó

**Affiliations:** Departament de Genètica, Microbiologia i Estadística, Facultat de Biologia and Insitut de Biomedicina (IBUB), Universitat de Barcelona, 08028 Barcelona, Catalonia, Spain; pllorens@ub.edu (P.L.-G.); carloscamilleri@ub.edu (C.C.-R.); mcorominas@ub.edu (M.C.)

**Keywords:** chromatin composition, chromatin organization, gene regulation, 3D genome structure, nuclear architecture

## Abstract

Eukaryotic genomes are packaged into high-order chromatin structures organized in discrete territories inside the cell nucleus, which is surrounded by the nuclear envelope acting as a barrier. This chromatin organization is complex and dynamic and, thus, determining the spatial and temporal distribution and folding of chromosomes within the nucleus is critical for understanding the role of chromatin topology in genome function. Primarily focusing on the regulation of gene expression, we review here how the genome of *Drosophila melanogaster* is organized into the cell nucleus, from small scale histone–DNA interactions to chromosome and lamina interactions in the nuclear space.

## 1. Introduction

In eukaryotic cells, nuclear organization refers to the spatial distribution of nuclear contents and components. The cell nucleus contains DNA, which is organized as multiple long linear molecules in a complex with a large variety of proteins, such as histones, to form chromatin and chromosomes. Thus, the eukaryotic genome is packaged into higher-order chromatin structures and organized in a manner that accommodates highly dynamic processes such as DNA replication, gene transcription, and DNA repair.

There are many different levels of nuclear organization and whether they affect gene function or just reflect this function is still unclear. Here, we focus on *Drosophila melanogaster*, a pre-eminent animal model system for genetic studies. Starting from basic DNA composition, we review what is currently known about the general structure of chromatin, chromosomes, and nuclear organization.

## 2. The *Drosophila* Genome

The first annotated whole genome sequence of the fruit fly *Drosophila* was published more than two decades ago, when it was estimated to contain around 13,600 genes [1,2]. The most recent version of the *Drosophila* genome identifies 13,969 protein-coding genes and 2545 long non-coding RNA genes, with a GC percentage of ~42% [3] (FlyBase r6.40, June 2021).

The *Drosophila* genome is divided into four chromosomes: the X and Y sex chromosomes, the autosomal chromosomes 2 and 3, and a tiny chromosome 4 containing no more than 100 genes and known as the “dot chromosome”. In *Drosophila*, like in other Diptera species, polytene chromosomes can be observed in the interphase nuclei of certain tissues such as the salivary glands. This highly specialized form of chromosomes develops by endoreduplication of the chromosomes of diploid nuclei, producing multiple chromatids of each chromosome. Polytene chromosomes have been very useful in cytogenetic studies due to their distinct patterns of bands and interbands showing different degrees of condensation, gene expression profiles, and protein composition [4,5].

## 3. Chromatin Composition and Structure

Eukaryotic DNA molecules, together with proteins and RNA, are packaged into a compact structure called chromatin. Different chromatin states have long been recognized, with chromatin classically divided into euchromatin (which decondenses regularly during the cell cycle, consists primarily of single-copy sequences, and is transcriptionally active) and heterochromatin (which is condensed throughout the cell cycle, consists mainly of repetitive sequences, and can silence gene expression) [6,7]. In *Drosophila*, heterochromatin comprises approximately a third of the genome and is organized primarily into pericentromeric and telomeric blocks [8]. Pericentromeric heterochromatin is mainly composed of repetitive sequences, including fragments of various transposable elements (TEs) and satellite DNAs (satDNA), which are large blocks of tandemly repeated DNA sequences [9]. Heterochromatin protein 1 (HP1a in *Drosophila*) is a conserved eukaryotic chromosomal protein that is associated with pericentromeric heterochromatin and mediates the concomitant gene silencing [10]. Despite its association with gene repression, it was reported recently that a significant part of the constitutive heterochromatin in *Drosophila* is, in fact, occupied by active genes [11]. Moreover, an RNAi screen conducted in flies revealed that heterochromatin is structurally complex and contains many dynamic smaller subdomains [12]. Beyond the binary classification of chromatin into euchromatin and heterochromatin, several groups have partitioned the *Drosophila* genome into different chromatin types or states based on a combinatorial signature of bound proteins, histone modifications and integrative analysis with other chromatin data [13] (discussed below).

### 3.1. Nucleosome Dynamics

The basic unit of chromatin is the nucleosome, which consists of an octamer composed of two copies of each of the core histones (H2A, H2B, H3, and H4) that is wrapped by 145–147 bp of DNA in a left-handed superhelical turn [14,15]. The core histones interact with DNA through the highly conserved histone-fold domain, while the N-terminal tail participates in nucleosome stabilization [15,16]. The different nucleosomes are separated by linker DNA and the resulting arrangement is an 11-nm chromatin fiber that resembles a beads-on-a-string structure [17]. Linker histones, such as H1, bind to DNA at the entry/exit site of the nucleosome, seal the structure, and protect an extra 20 bp of DNA [18,19,20]. The resulting structure is called a “chromatosome” [21]. Several studies have addressed the role of H1 in chromatin folding, showing that H1 promotes and stabilizes the compaction of nucleosomes into a 30-nm chromatin fiber [22,23]. However, the 30-nm fiber is only observed as short fragments in vivo, since nucleosomes are found in clutches of various sizes separated by nucleosome-depleted regions [24].

The organization of nucleosomes varies across the genome and plays a central role in controlling DNA accessibility. Nucleosome-depleted regions are characteristic of active chromatin sites and adjacent regions show a regular placement of nucleosomes. More irregular positions are commonly found elsewhere [25,26,27]. The determinants of nucleosome positioning were defined some years ago as a combination of DNA sequences, ATP-dependent chromatin remodeling enzymes, transcription factors (TFs), and elongating RNA polymerase II (RNAPII) [28]. A comparative analysis between *Drosophila* cell lines identified genomic regions that exhibited cell line-specific nucleosome enrichment or depletion. The same study revealed that nucleosomes were positioned in accordance with previously known DNA–nucleosome interactions, with helically repeating A/T di-nucleotide pairs arranged within nucleosomal DNA and AT-rich pentamers generally excluded from nucleosomal DNA [29].

Nucleosomes are highly dynamic structures and the partial unwrapping of DNA from the octamer leads to exposure of the different regions for protein recognition [30,31]. Nucleosome dynamics are controlled by a complex cooperation between different histones, histone post-translational modifications, nucleosome occupancy and positioning [32,33]. In recent years, the development of genome-wide mapping approaches has provided a vast amount of information regarding the genomic location of chromatin-associated proteins, such as histones, as well as of several of their modifications. A large number of these modifications have been implicated in the regulation of gene expression, as we will discuss below.

### 3.2. Core Histones and Their Variants

*Drosophila* contains five canonical histones (H1, H2A, H2B, H3, and H4), usually referred to as replication-coupled histones since they are mostly incorporated during DNA replication. The canonical histone genes are clustered into a highly repeated unit that contains one copy of each gene, although H4 is also encoded by another gene outside the cluster (*H4r*) [34].

In addition to the canonical histones, the fly genome encodes four histone variants (BigH1, H2Av, H3.3, and cenH3), with H2B and H4 being the only histones lacking variants. Histone variants confer different structural properties and carry out specialized functions in numerous processes [35]. The unique *Drosophila* H2A variant, H2Av, combines the features of the H2A.X and H2A.Z eukaryotic variants and has been linked to transcription, DNA repair, and heterochromatin [36]. H2Av may also have a role in chromosome organization, since the depletion of the machinery responsible for its incorporation results in the alteration of chromosome structure in salivary glands and S2 cells [37,38]. H2Av is broadly distributed in the *Drosophila* genome and nucleosomes with H2Av are particularly enriched downstream of the transcription start site (TSS) of active genes, which correlates positively with transcription levels [25,39]. It has been proposed that H2Av may facilitate the progression of RNAPII, since a reduction of H2Av levels results in an increase in RNAPII stalling [40]. In addition, H2Av has been implicated in gene silencing through the Polycomb group (PcG) of proteins. H2Av seems to participate in Polycomb (Pc) recruitment since Pc sites are lost in polytene chromosomes from H2Av mutants [41]. Moreover, H2Av is also found in heterochromatin [25,42] and HP1a binding may depend on the presence of H2Av [41].

The H3 replacement variant H3.3 is encoded by two genes, H3.3A and H3.3B, which are ubiquitously expressed throughout all the tissues and developmental stages [43]. H3.3 is usually enriched in active chromatin, such as active promoters and gene bodies of transcribed genes [44,45], and in sites with high nucleosome turnover rates [46]. However, clonally removing H3.3 in cells from the *Drosophila* wing disc does not affect gene expression [47]. Furthermore, H3.3 null mutant flies are viable and show no phenotypic alterations, except for infertility [47]. Interestingly, regions of H3.3 enrichment are generally depleted of H1 and knocking down H3.3 leads to increased H1 association at these sites [48].

### 3.3. Linker Histones

*Drosophila* contains only one somatic H1 [49] and one embryonic and germline-specific variant, BigH1 [50]. H1 is broadly distributed throughout the genome [48,51]. Its loss results in the misexpression of only a small subset of genes [52,53], mainly those located in heterochromatin [54,55]. The changes in gene expression include the upregulation of TEs [52,53,56]. In mammals, linker histones are also widely distributed [57,58,59,60] and, similar to flies, only a small number of genes are affected in triple knock-out mouse embryonic stem cells (ESCs) with 50% total H1 depletion [61] or in human cells with reduced H1 levels [59,62,63]. Moreover, these changes in gene expression mainly affect heterochromatic regions [57,59,62,64].

BigH1 is expressed in both male and female germlines with similar expression patterns [50,65,66]. While the function in the female germline is not known, in males, BigH1 is important for germ stem cell maintenance and spermatocyte differentiation [65]. BigH1 is retained in precellular embryos, where it is important in maintaining the silencing of the zygotic genome [50]. Recent studies have focused on the different properties of the two linker histones and have found that BigH1 has a greater repression ability than the somatic H1 due to its higher content in acidic residues [66]. Moreover, nucleosomes containing BigH1 are more stable, but show higher mobility than H1-nucleosomes [67].

## 4. Covalent Modifications of Chromatin

Covalent modifications play an essential role in the nucleosome–nucleosome interactions that dictate chromatin folding and dynamics. These modifications, occurring both in DNA and histones, have the potential to form a complex combinatorial regulatory system and are fundamental in regulating all processes that use DNA as a template, such as transcription, DNA repair, and replication.

### 4.1. DNA Methylation

Methylation of the carbon C5 of cytosine to form 5-methylcytosine (5mC) is probably the best-known modification of DNA in eukaryotes. Despite the general role 5mC plays in the repression of vertebrate gene expression, the situation may be different for invertebrates. There is evidence in favor of cytosine C5 methylation in *Drosophila*, although its source is still elusive [68]. Although 5mC is rare, methylation on N6 adenine (6mA) seems prevalent in *Caenorhabditis elegans* and *Drosophila* [69,70]. Recent studies have confirmed that NMAD-1 (MT-A70 family) and DMAD (DNA 6mA demethylase, TET ortholog) are 6mA demethylases in *C. elegans* and *Drosophila* respectively [71]. The *Drosophila* DMAD regulates 6mA levels during embryo development and oogenesis. DNA immunoprecipitation studies in ovaries from DMAD mutants show that 6mA is enriched in transposon regions and seems to promote their expression [70]. More recently, He and coworkers (2019) have shown that, in *Drosophila* embryos, 6 mA is found not only in transposon regions but also in zygotic genes. This modification can be read by the TF Jumu, which controls the activation of the zygotic genome (ZGA) through the regulation of *zelda*, among other genes [72].

### 4.2. Histone Modifications

Several residues in histones are susceptible to modification. The most well studied modifications are the ones occurring at the N-terminal tails of the core histones, although modifications of the globular domains have been gaining more attention [73,74]. The N-terminal tails of the core histones protrude outside the nucleosome and contact with adjacent nucleosomes. Thus, modifications in this region can directly affect nucleosome–nucleosome interactions and alter the chromatin structure. This is the case for H4K16 and H4K20. While acetylation of H4K16 reduces the level of compaction in vitro [75,76], the di- and tri-methylation of H4K20 has the opposite effect and enhances chromatin condensation in vitro [77]. However, modifications at the N-terminal tails also mediate the recruitment of effector proteins [78]. Recognition of these modifications is achieved through specialized domains present in “reader” proteins, which can be remodeling complexes, other modifying enzymes or scaffolds of the transcription machinery [78]. Some of the effector proteins and the enzymes that catalyze or remove these modifications are part of the PcG or Trithorax group (TrxG), which are chromatin-modifying complexes implicated in the maintenance of repressed or active gene expression states [79].

There are some modifications that correlate with gene activity [80]. The clearest example is the acetylation of histones H3 and H4, which is associated with active transcription. Acetylation of lysine (K) partially neutralizes the positive charge of histones, thus weakening the interaction between histones and DNA [81]. Methylation is, by far, more complex, and its correlation with gene expression depends not only on which amino acid is modified, but also on the degree of methylation (mono, di or tri). The hypothesis of the “histone code” proposes that the combination of different modifications is important in regulating gene expression and other DNA processes [82]. Recent studies, however, challenge the instructive role of histone modifications and suggest that those modifications traditionally associated with active genes do not directly trigger transcriptional activation [83]. Indeed, typical histone modifications only have a few roles in regulating transcription [84]. A study using different developmental time points showed that the transcription of genes temporally regulated during fly and worm development occurred in the absence of canonically active histone modifications [85]. Similarly, another study using genetic approaches and mutant derivatives found that transcriptional regulation can occur in the complete absence of H3K4 methylation [86]. Likewise, the depletion of H3K27ac in mouse ESCs does not alter chromatin accessibility or transcription, indicating that this modification is dispensable for enhancer activity in mouse ESCs [87]. These modifications might instead be necessary for sustained transcription, since there is a correlation between the amount of these modifications and the stability of expression [85].

Another example of transcription without the typical active histone modifications is the first wave of transcription of the ZGA, which is characterized by the enrichment of H4K8ac, H3K18ac, and H3K27ac in active genes [88]. Other modifications commonly associated with active genes, such as H3K4me3, H3K9ac, H3K36me3, and H3K4me1, do not become enriched until mitotic cycle 14 [88,89], indicating again that these modifications are not required for transcriptional competence, at least during the first wave of transcription. Recently, H3K14ac was shown to be important for the transcription of active genes that lack H3K9ac, H3K27ac, and H3K4me3 during *Drosophila* embryo development and in imaginal wing discs [90]. Moreover, modifications in the globular domain may have a role in regulating gene expression and other processes, since mutations of the H3 residues K56, K115, K122, T80, and T118 induce lethality at different developmental stages in *Drosophila* [91].

## 5. Functional Organization of the Genome

### 5.1. Segmentation of the Genome into Chromatin States

As mentioned above, several groups have attempted to classify chromatin into different types. Filion et al. (2010) proposed the segmentation of the genome of *Drosophila* cells into five types of chromatin based on genome-wide binding maps of selected chromatin components, with histone modification profiles used for independent validation (Figure 1A). These chromatin states, labeled as GREEN, BLUE, BLACK, RED, and YELLOW, are distributed throughout the genome in discrete domains with a length usually ranging from ~1 to 52 kb. The GREEN and BLUE chromatin types correspond to classic heterochromatic regions and are characterized by the binding of HP1 and HP1-interacting proteins or of PcG proteins, respectively. While GREEN chromatin is marked with H3K9me2 and usually corresponds to pericentromeric regions, BLUE chromatin is highly enriched in H3K27me3 and developmentally regulated genes. The last type of silent chromatin is BLACK chromatin, which covers about half of the genome and tends to have longer domains. BLACK chromatin is thus the predominant type and is characterized by being poor in genes and producing very low levels of mRNA. Interestingly, the genes within this type of chromatin can become active in specific cell types or tissues, suggesting that BLACK domains can be remodeled into a different chromatin type during development. Moreover, some of the proteins that mark this type of chromatin are histone H1, auroraB (AurB), Suppressor of Underreplication (SUUR), the AT-hook protein D1, and lamin (LAM), indicating a role of the nuclear lamina in the regulation of BLACK chromatin. The two remaining types of chromatin, RED and YELLOW, correspond to active regions and are characterized by the extensive binding of histone deacetylases (HDACs) and ASH2, as well as enrichment in H3K4me2, H3K79me3, and RNAPII. RED chromatin is also bound by several other proteins, including TFs such as the GAGA factor (GAF) and Jun-related antigen (JRA). Instead, YELLOW chromatin is uniquely marked by H3K36me3 and its reader protein MRG15. The genes located in RED and YELLOW chromatin are also different: the genes in RED chromatin have specific expression patterns and functions, such as signaling and TF activity, whereas those in YELLOW chromatin are ubiquitously expressed and have more general functions, such as DNA repair and metabolism [13].

Following a similar approach, Kharchenko et al. (2011) used histone modifications to determine nine distinct chromatin states in the *Drosophila* genome (Figure 1B). To functionally characterize these states, the authors integrate data from non-histone proteins, chromosome accessibility, transcription analyses, and short RNA production. In the 9-state model, transcriptionally active regions fall into more than two chromatin states, some of which can be observed at different regions of a particular gene. State 1 (red) is characterized by the enrichment of H3K4me3/me2 and H3K9ac, and is found at active promoters and TSSs. State 2 (purple) contains high levels of H3K36me3, an elongation mark enriched towards the 3′ end of the genes. State 3 (brown) is usually found within intronic regions and is enriched in H3K27ac, H3K4me1 and H3K18ac. Open chromatin regions are mainly associated with states 1 and 3, which are bound by different components of chromatin remodeling factors, such as NURF301 and MRG15 in the case of state 1, or SPT16 and dMI-2 in state 3. The authors propose a regulatory role for the state 3 domains, since they are also enriched in dCBP/p300 and almost half of them are bound by GAF and developmental TFs. State 4 (coral) resembles state 3, but lacks H3K27ac and is also marked by the presence of H3K36me1. Chromosome X is particularly enriched in state 5 (green), which is defined by high levels of H4K16ac and the modifications also present in state 2, probably reflecting a distinct mechanism of transcriptional regulation required for dosage compensation in male cells. State 6 (dark grey) is enriched in H3K27me3 and corresponds to PcG-repressed regions. Heterochromatic regions are depicted by state 7 (dark blue) and 8 (light blue), which are characterized by an enrichment in H3K9me2/m3, although levels are higher in state 7. Finally, the authors consider a last chromatin state characterized by the presence of low levels of the histone modifications considered in the study, the state 9 (light grey) [92].

These different types of chromatin have been related to physical domains of chromosome folding. For example, PcG-bound chromatin (BLUE chromatin or state 6) has been shown to form small subnuclear structures called PcG bodies or PcG-repressed domains [93,94]. As we discuss below, there is a strong link between chromatin activity and chromosome architecture.

### 5.2. 3D Organization of the Genome

Technical advances such as the high-throughput derivative of chromosome conformation capture (Hi-C) have enabled the analysis of the three-dimensional architecture of genomes (Figure 2A) [95,96,97]. Chromatin interaction maps generated by Hi-C assays in *Drosophila*, mice, and humans have revealed that the genome is composed of several layers of structure that are organized in a hierarchical manner [98,99,100,101]. At the higher level of genome topology, chromatin is partitioned into two multimegabase compartments with distinct transcriptional activity: an active compartment (A) that is dense in expressed genes and correlates with histone modifications generally associated with active transcription, and an inactive compartment (B) that is gene-poor and heterochromatic [97,100]. The genome is further organized into sub-megabase domains called topologically associating domains (TADs), which are defined by a higher interaction frequency within the region than with those located outside of the TAD [98,102,103]. In mammals, TADs are likely to be formed from the active extrusion of chromatin loops mediated by the cohesin complex [104,105], and are insulated at the borders by the architectural chromatin protein CCCTC-binding factor (CTCF) [103,106]. Depletion of cohesin in mammalian cells results in the loss of the majority of TADs [107], while deleting CTCF sites or inverting their orientation reduces TAD insulation and facilitates crosstalk between adjacent TADs [108,109,110].

In *Drosophila,* high-resolution Hi-C data from Rowley et al. (2017) suggested that TADs and compartments occur at a much lower scale than previously proposed. According to that study, the main topological features are compartmental domains, which represent small discrete regions of ~10 kb that preferentially interact within themselves and correlate with transcriptional activity states [111], thus indicating that *Drosophila* TADs actually correspond to smaller A/B compartments. This is consistent with the segmentation of the fly genome into the domains of the particular chromatin types mentioned above [13]. Thus, A compartmental domains would correspond to RED and YELLOW chromatin, while B compartmental domains would include BLUE, GREEN and BLACK chromatin types. These compartmental domains can be accurately modeled using only transcriptional data, while both CTCF and transcription-based simulations are required to generate an accurate human Hi-C map [111]. Significantly, the experimental inhibition of transcription in *Drosophila* cells by triptolide or heat shock results in a decrease in domain architecture that is more pronounced when RNAPII binding is also depleted [111]. In another study from Hug et al. (2017), Hi-C maps generated during fly embryogenesis showed that most TADs are formed concomitantly with the start of transcription. Inhibiting transcription before ZGA results in reduced contact density within domains and a significant loss of interdomain insulation, although TADs are not entirely eliminated [112].

Unlike in mammals, the majority of *Drosophila* domain borders coincide with active promoters or active gene minidomains rather than with CTCF sites [111,113]. Accordingly, domain boundaries are mostly unaffected by the loss of CTCF, as has been observed in *Drosophila* neurons [114]. CTCF mutant flies can progress through embryogenesis and larval stages, although they display strong homeotic defects [115]. Instead, many other insulator proteins have been described in *Drosophila*; however, their role in shaping chromatin domains is still unclear [116]. In recent studies using Hi-C data from *Drosophila* cell lines, significantly enriched DNA motifs have been identified at domain boundaries, including the motifs for the Suppressor of Hairy wing [Su(Hw)], the Boundary Element Associated Factor (BEAF-32), and the Motif-1 binding protein (M1BP) [113,117]. Interestingly, loss of BEAF-32 has no major effect on chromosome structure, while depletion of M1BP results in cell arrest and dramatic genome reorganization [113]. Another study revealed an essential role for the pioneer TF Zelda in establishing insulation at domain borders, at least partially, during embryogenesis [112]. That study showed that Zelda-depleted embryos display major chromatin conformation defects, although they are mostly locus-specific, i.e., at former Zelda-bound domain boundaries [112]. Furthermore, Zelda mutant embryos show loss of long-range gene interactions [118], suggesting a role of this pioneer factor in chromatin looping during *Drosophila* embryogenesis. Significantly, Zelda-dependent sites that fail to interact in mutant embryos are associated with the specific depletion of RNAPII binding [118]. Overall, these results suggest that compartmentalization in the fly genome is mostly driven by transcriptional activity and/or RNAPII presence, while CTCF looping is also required for domain formation in mammals. A recent study has proposed that mammalian TADs are subdivided into chromatin nanodomains of similar size to *Drosophila* TADs. These nanodomains, like *Drosophila* TADs, are mostly unaffected by CTCF or cohesin depletion [119].

The recent application of super-resolution microscopy coupled to DNA labeling has revealed that *Drosophila* TADs correlate well with epigenetic states, usually classified into transcriptionally active (associated with H3K4me3, H3K36me3 and acetylated histones), PcG-repressed (enriched in PcG proteins and H3K27me3), and inactive (devoid of specific marks) (Figure 2B). Using a combination of fluorescent in situ hybridization (FISH) and 3D-structured illumination microscopy (3D-SIM), Szabo and colleagues (2018) showed that the chromatin fiber is segmented into globular domains defined as nanocompartments [120]. These structures correspond to PcG or inactive TADs, and are interspersed by regions of less condensed active chromatin [120]. Similarly, 3D stochastic optical reconstruction microscopy (3D-STORM) of labeled chromatin revealed distinct spatial organization for each chromatin state, where PcG-repressed chromatin forms small compact domains that are distinct from the transcriptionally inactive regions [94]. Interestingly, active and inactive domains show less condensation and partial intermixing with each other, while PcG-repressed domains exhibit more compact packaging and a stronger tendency to spatially exclude neighboring active chromatin [94]. These PcG domains depend on Polycomb repressive complex 1 (PRC1), since knockdown of the Polyhomeotic (Ph) component of PRC1 leads to domain loss and an increased expression of PcG-target genes [94]. Similarly, genome editing of Polycomb responsive elements (PRE) caused the aberrant formation of TADs and a decreased silencing during *Drosophila* embryogenesis [118]. In mammals, PcG-bound chromatin also forms small dense domains in which developmentally regulated genes targeted by PRC1 are corepressed [121]. These domains are functionally lost upon the activation of PRC1-target genes or upon the PRC1 unbinding of target genes during cell fate specification [121]. The function of Ph in domain formation is also conserved for the mammalian ortholog Phc1, since knocking out Phc1 in mouse ESCs leads to domain loss and subsequent gene de-repression [121]. A recent study used Optical Reconstruction of Chromatin Architecture (ORCA), a method that combines high resolution microscopy with Oligopaint and RNA-FISH, to observe the 3D chromatin organization of the bithorax-complex (BX-C) inside the nuclei of individual cells from *Drosophila* embryos. The study revealed segment-specific organization of the BX-C and identified TADs exclusive to each body segment [122]. The boundaries of these segment-specific TADs often coincide with changes in chromatin states, such as at the edge of a H3K27me3 domain or at the border between transcriptionally active and inactive regions. However, the same study found adjacent TADs with the same epigenetic states that were independent of H3K27me3. The borders of these TADs might be determined by cohesins and CTCF, since the deletion of border regions marked by these proteins results in TADs fusion. Nevertheless, these deletions also include active genes, which could influence the structure of TADs [122]. Finally, another study showed that the boundaries of most Polycomb domains in *Drosophila* consist of active regions and that actively transcribed genes can stop the spreading of H3K27me3, since ectopic addition of a transcriptional terminator caused the extension of the PcG domain [123].

The results discussed above indicate that the fly genome is divided into functional domains in which regulatory contacts are spatially restricted. However, this model requires TADs to be physical units when they could instead represent statistical frequencies of chromatin interactions within cell populations [124]. Single-cell Hi-C (scHi-C) has been recently introduced to overcome the limitations of bulk Hi-C, but have shown contrasting results in this respect. The contact maps generated for single *Drosophila* nuclei bore striking resemblance to the TAD profile in bulk Hi-C, with over 40% of TAD boundaries conserved between individual cells [125]. However, the long-range interactions within and between TADs were very heterogeneous, suggesting substantial stochasticity in the folding of chromatin into domains [125]. Another study using human cell lines showed high cell-to-cell variations in contact patterns, where single-cell contact clusters did not match population TADs [126]. These scHi-C maps averaged into TADs when they were pooled together, suggesting that TADs merely reflect the tendencies of measured interactions within a population of cells [126]. Using 3D-SIM and DNA-labeling in *Drosophila*, contact probabilities within TADs were shown to be higher for all cells compared to contacts between neighboring TADs [127]. This preferent confinement of interactions within domains indicates the presence of TADs in individual cells. Furthermore, Cattoni and colleagues (2017) showed that active and repressive chromatin form discrete nanocompartments at the single-cell level, whereas stable looping between TAD borders is infrequent. This suggests that TAD assembly in flies is not the result of long-term stable interactions, but rather can be explained by the stochastic contacts between regions of similar chromatin types [127]. In agreement with this, Szabo and colleagues (2018) observed that TADs are generally consistent between cells, despite variable intra- and interdomain contacts, further suggesting that TADs are true physical entities of the fly genome [120]. Likewise, super-resolution microscopy of mouse ESCs showed significant variability at the TAD scale between individual cells, although chromatin contacts were more frequent within TADs than between adjacent TADs [119]. Further evidence in favor of TADs is the observation that bands from *Drosophila* polytene chromosomes strongly correlate with TADs, whereas the decondensed polytene regions, named interbands, mostly correspond to inter-TADs or TAD borders [128,129]. This finding reveals a direct relationship between sequencing-inferred chromatin structure and chromosome condensation observed by light microscopy [128,129], suggesting that TADs are a stable and conserved unit of chromosome folding.

### 5.3. Regulation of Gene Expression by Chromatin Organization

While there is an increasing amount of evidence showing that transcriptional activity plays a prominent role in chromatin organization [111,130,131,132], it is still rather controversial how—and if—genome topology modulates gene expression [133]. In a recent study, the simultaneous analysis of enhancer–promoter interactions and transcription in *Drosophila* embryos showed that the sustained proximity of the enhancer to its target gene was required for activation [131]. Moreover, the *Drosophila* Hox gene clusters Antennapedia and Bithorax, which are separated by 10 Mb in the linear genome, are thought to be corepressed by a PcG-mediated physical interaction, with mutations in one complex resulting in the de-repression of the genes in the other complex [93]. Interestingly, the absence of either one of the two PRC1 subunits Ph or Pc affects chromatin organization prior to ectopic Hox gene transcription, suggesting that PRC1 maintains gene silencing by compacting chromatin into domains [134]. In agreement with these results, disruption of the TAD structure has been found to cause misexpression in some cases. For instance, the deletion or inversion of the TAD boundary at the *Epha4* locus causes ectopic interactions between promoters and enhancers, leading to aberrant gene expression and a pathogenic phenotype in mammalian limbs [135]. Furthermore, deletion of domain boundaries in the *Notch* locus of *Drosophila* results in TAD fusion and transcriptional changes, together with a decreased binding of RNAPII [136].

However, there is also significant evidence questioning the role of the chromatin structure modulating gene expression [133,137]. In *Drosophila*, the analysis of gene expression in balancers (highly rearranged chromosomes) showed that disruption of TADs has little effect on gene activity and only a subset of genes is sensitive to structural alterations [138]. Similarly, a recent study using *Drosophila* dorsoventral patterning as a model system indicated that genome topology and transcription are independent [139]. Following fly embryonic development, the study demonstrated that chromatin conformation is generally maintained across cell types, even if transcriptional profiles change dramatically [139]. Likewise, single-cell spatial genomics in *Drosophila* embryos revealed that developmentally relevant enhancer–promoter interactions appear before TAD formation and remain invariant during cell fate specification [140]. Finally, sonic hedgehog (*Shh*) gene expression during mouse limb development was shown to be robust to perturbations in the TAD structure and was only altered when the Shh limb enhancers were deleted [141]. Altogether, these results suggest that chromatin organization can play a role in regulating gene expression, but it is most likely to be one of many modulators and not the main regulatory factor.

### 5.4. Nuclear Lamina and Pericentromeric Heterochromatin

The functional organization of the genome is also influenced by the existing contacts between the chromatin and the nuclear lamina (NL) (Figure 1A) [142]. The NL is a dense meshwork composed of A- and B-type lamins and lamin-associated proteins that, together with the outer and inner membranes and the nuclear pore complexes, form the nuclear envelope. The NL functions as a support for multiple chromatin anchoring sites and it is suggested that NL proteins tether heterochromatin to the nuclear periphery. In fact, *Drosophila* genes that interact with B-type lamin are transcriptionally silent and late replicating [143]. During the differentiation of mouse ESCs, genes that move away from the lamina are activated, whereas the others remain inactive and become activated in the next differentiation step [144].

Lamin B receptor, located at the inner nuclear membrane, provides a tethering mechanism for heterochromatin by binding to HP1a, which is preferentially located in the H3K9me3-rich heterochromatic region [145,146,147]. Hi-C data have also confirmed that the inactive chromatin compartment is strongly enriched in NL contacts [148,149]. In addition, transcriptional repressors, such as HDACs, are known to bind to lamina proteins [150,151,152], raising the question of whether the contacts with the NL make the chromatin in lamina-associated domains (LADs) compact and inactive. Evidence indicates that the NL contacts can play a role in gene repression. In vitro experiments on *Drosophila* cell lines have demonstrated that knocking down lamin proteins decreases the compactness of inactive chromatin domains, increases the accessibility of the promoters located in heterochromatic regions, enhances the levels of histone H3 acetylation, and increases gene expression [144,153,154,155]. Similarly, in mouse ESCs, the expression of reporter genes inserted into LADs is remarkably lower when compared to inter-LADs, which is only partially explained by chromatin compaction, therefore pointing towards a repressive environment near the NL where transcription is less permissive [156]. Additionally, several studies on cell differentiation have revealed that the activation of tissue-specific gene expression is associated with translocations of loci from the NL to the nuclear interior [144,149,157,158]. For instance, the ectopic release of cardiac genes from the nuclear periphery in mouse ESCs leads to a premature myogenesis, providing a clear example of the relevance of gene positioning in the nuclear periphery during organogenesis [159]. Together, these findings indicate that the NL is essential for the positioning of the heterochromatin in the nuclear periphery, establishing a transcriptionally repressed domain near the nuclear envelope and a transcriptionally active domain in the nuclear interior [160].

Apart from the heterochromatin contacts with the NL, interactions between different heterochromatic regions, most likely mediated by the affinity between repetitive elements [161,162] or heterochromatin-associated proteins [163,164], could be essential for heterochromatin compartmentalization within the nucleus [165]. Contacts between pericentromeric heterochromatin (PCH) regions, which are located near the centromere and are enriched in HP1a and repetitive elements [166,167], have been proposed to contribute to the global genome architecture in the nucleus [168]. In *Drosophila*, the PCH regions from the four chromosomes cluster in the 3D nucleus, forming membraneless structures called chromocenters that were first described in the 1970s [169,170,171]. Although recent studies have found HP1a to be crucial for PCH clustering in *Drosophila* embryos [172], the mechanism driving this clustering in differentiated cells in mice does not depend on HP1a, indicating an alternative mechanism for PCH coalescence [173]. However, PCH contacts are not only restricted to other PCH regions. It was recently found that PCH can interact with euchromatic regions enriched in H3K9me2/3. Genes located in PCH-contacting euchromatin show higher expression than those located in euchromatin not contacting PCH [168]. Since HP1a is the reader of H3K9me2/3, the most probable scenario is that PCH-euchromatic interactions are driven by HP1a.

### 5.5. Nuclear Pore Complexes

Besides the nuclear lamina, chromatin can also interact with the nuclear pore complexes (NPCs) located in the nuclear envelope and whose canonical function is to mediate nucleocytoplasmic transport. NPCs are composed of multiple nucleoporins (Nups), for which ~30 different subtypes have been described [174]. Depending on the Nup subtype, the association to the NPC can be stable or dynamic, as certain Nups can be found freely within the nucleoplasm. In *Drosophila*, as well as in mammals, chromatin binding maps show differential binding preferences for different Nup subtypes. For instance, Nup153 and Megator (Mtor) are known to bind to the fly genome in continuous domains enriched in marks of active transcription [175]. In fact, the depletion of Nup153 results in the altered expression of thousands of genes, suggesting a role in transcriptional regulation [175]. However, not all Nup subtypes bind preferentially to active regions. An interesting example is the antagonistic binding preferences of core nuclear pore proteins Nup93 and Nup107 in *Drosophila* cells; while Nup107 is mainly found at active genes, Nup93 is located preferentially at repressed chromatin regions, bound by PcG proteins [176]. Indeed, the depletion of Nup93 leads to de-clustering of distant Polycomb regions that previously coalesced, and to de-repression of PcG target genes [176]. These findings point towards Nup93 as a key mediator of long-range Polycomb interactions, as well as a necessary factor for proper silencing of PcG-associated target genes.

Dynamic Nups can also interact with the genome within the nucleoplasm, outside the NPC. This is the case of Nup98, which binds genes both at the NPC and within the nucleoplasm [177,178]. The association involving nucleoplasmic Nup98 correlates with a higher degree of target gene activation compared to NPC-bound Nup98. The expression of these target genes depends on the levels of Nup98, as their expression increases upon the ectopic overexpression of Nup98, and decreases upon Nup98 depletion [178]. In line with these findings, the interacting partners of Nup98 include several proteins implicated in gene activation, including Thritorax (Trx), suggesting a role of Nup98 in maintaining active gene expression [179].

NPCs also seem to play a role in genome architecture. Multiple studies have demonstrated that NPC components contribute to the formation of long-range genomic contacts [180,181,182]. In *Drosophila*, as well as in mammals, several Nups are found targeting a subset of promoters and enhancers. These contacts are preferentially located in NPC-bound Nups rather than nucleoplasmic Nups, and occur regardless of the transcriptional state of the genes, suggesting that NPCs can target silent or poised genes and enhancers [180,181]. Particularly, Nup98 was found to bind promoters and enhancers of ecdysone-inducible genes and, upon its depletion, the enhancer–promoter loops induced by ecdysone were destabilized [180]. Moreover, NPC-bound Nup98 was also observed to bind at a subset of TAD boundary regions, and was identified as a physical interactor to insulator proteins such as CTCF [113]. Altogether, these findings suggest that NPCs not only bind to specific regions in the genome, but may also participate in the formation or maintenance of promoter-enhancer loops as well as in the formation of TAD borders.

## 6. Liquid–Liquid Phase Separation

Functional compartmentalization of the cell nucleus plays an important role in the regulation of genome activity. Recent evidence suggests that liquid–liquid phase separation (LLPS) underlies the formation of membraneless compartments in the nucleus [183]. These membraneless compartments are formed as a result of distinct biochemical properties that segregate macromolecules into a concentrated liquid phase and a dilute phase, referred to as condensates [184]. This non-covalent form of fluid compartmentalization is triggered by weak multivalent interactions between proteins, RNA, and DNA. Specifically, proteins are thought to mediate phase separation through interactions between domains that are called low-complexity domains (LDRs) or intrinsically disordered domains (IDRs) [184,185]. Recent studies suggest that chromatin compartments might be formed and organized through LLPS (Figure 2C). In *Drosophila*, HP1a undergoes liquid phase separation in vitro and can form condensates suggested to facilitate heterochromatin formation in early embryos [164]. Indeed, partial knock-down of HP1a causes major alterations in chromatin organization in *Drosophila* embryos, such as a reduced contact frequency within heterochromatic regions and increased crosstalk between active and inactive domains [172]. HP1a depletion in differentiated cells does not affect genome structure, suggesting that HP1a is required for establishing 3D structure in early embryos, but not for the maintenance of compartmentalization during cell differentiation [172]. The human counterpart, HP1α, is also capable of compacting heterochromatin through phase separation in an in vitro model system [163]. However, a recent study using mouse fibroblasts showed that HP1α has a weak capacity to form liquid droplets in living cells and that the compaction and maintenance of heterochromatin foci occur independently of HP1α [173].

Other proteins have been suggested to facilitate chromatin condensation through LLPS, including histones and transcriptional regulators. The linker histone H1 was recently shown to promote the phase separation of reconstituted chromatin into denser and less dynamic droplets [186]. On the other hand, acetylation of histone tails gradually reduces chromatin droplet density until its dissolution. This acetylated chromatin, which is unable to undergo phase separation by itself, undergoes LLPS after the addition of multi-bromodomain proteins in vitro [186]. Similarly, the Chromobox 2 (CBX2) component of PRC1 was recently proposed to mediate the phase separation of PcG-repressed chromatin in mice. In an in vitro model, reconstituted PRC1 can undergo phase separation into droplets, but fails to do so when CBX2 is mutated in its LDR [187]. The same mutations in CBX2 had been previously shown to decrease chromatin compaction and transcriptional repression in mouse cells [188]. Finally, superenhancers, which are clusters of enhancers that are able to recruit high levels of transcriptional regulators and can strongly activate gene expression [189,190], are also thought to assemble through LLPS [191]. Indeed, recent studies suggest that TFs and coactivators can form phase-separated condensates, in which the transcription machinery is highly concentrated and facilitates the expression of genes in both flies and mammals [191,192,193]. Interestingly, the carboxy-terminal domain of RNAPII, an LDR, can undergo phase separation in vitro even in the absence of other proteins [194]. Moreover, the activation domains of several TFs can form liquid droplets in vitro with the coactivator mediator subunit MED1, which results in gene activation [195]. Altogether, the evidence suggests that chromatin has an intrinsic capacity to undergo phase separation into functionally distinct, but physically adjacent domains. However, the precise contribution of LLPS to the formation of chromatin domains and genome function is still unclear.

## 7. Conclusions and Future Perspectives

In this review, we aimed to highlight what is known about the genome of *Drosophila melanogaster* regarding chromatin composition and 3D organization, mostly related to the regulation of gene expression. As we have discussed, the role that chromatin modifications play in transcription is still unclear, and new studies are now focusing on the effects of local chromatin environment and genome folding. In spite of several open questions, the 3D organization of the genome in nuclear compartments is increasingly recognized as a major feature of gene regulation. How these nuclear compartments are formed and what is their importance regarding gene function remains to be elucidated.

Over the last decade, numerous studies have used chromosome conformation capture techniques to analyze the 3D architecture of genomes [95]. In *Drosophila*, these studies have revealed that the genome is organized into physical domains of particular chromatin states named TADs. However, Hi-C experiments reflect averaged information coming from cell populations, and thus cannot account for individual heterogeneity between cells, cell-types or tissues. To overcome this limitation, new methodologies have been developed, including scHi-C [126] and microscopy-based techniques, which couple super-resolution imaging with DNA labeling [120,122,196,197]. These emerging microscopy methods not only enable the visualization of the 3D genome organization at the single-cell level, but also maintain the spatial information within tissues or organisms. Furthermore, they allow the incorporation of RNA probes to distinguish cell types and to link chromatin structure to gene expression [122,196,197]. Other technical approaches to study the relationship between genome organization and gene expression include gene silencing by RNA interference (RNAi) and genomic engineering with CRISPR/Cas9 technology [198]. In *Drosophila*, CRISPR/Cas9 screens have recently been used to identify factors involved in genome architecture, including chromatin-binding proteins and functional cis-regulatory elements, such as enhancers and boundary elements or insulators [198]. Furthermore, the development of new methods based on a nuclease-dead Cas9 (dCas9) coupled with live imaging is allowing researchers to study the 4D genome, changes in 3D chromatin structure over time [199]. In the CRISPR/dCas9 system, there is specific recruitment of a fluorescently labelled dCas9 to the genomic region of interest, which allows the tracking of the contact and folding dynamics of this region by live microscopy [199].

Sequencing-based technologies rely on the available version of the genome and, in the case of *Drosophila*, the assembly of heterochromatic regions is still incomplete. New technologies based on long-read sequencing will improve the current genome assembly and thus increase the quality of Hi-C maps. In the field of microscopy, one of the major limitations is the number of loci that can be probed simultaneously. Recent advances including ORCA [122], Hi-M [197], and OligoFISSEQ [200] allow the sequential labeling and imaging of multiple genomic sequences, but still require long acquisition times and are very expensive. Altogether, the development of these techniques is encouraging, and further optimization will allow us to resolve the chromatin dynamics of whole genomes and to uncover the degree of heterogeneity inside cell populations.

## Figures and Tables

**Figure 1 cells-10-02362-f001:**
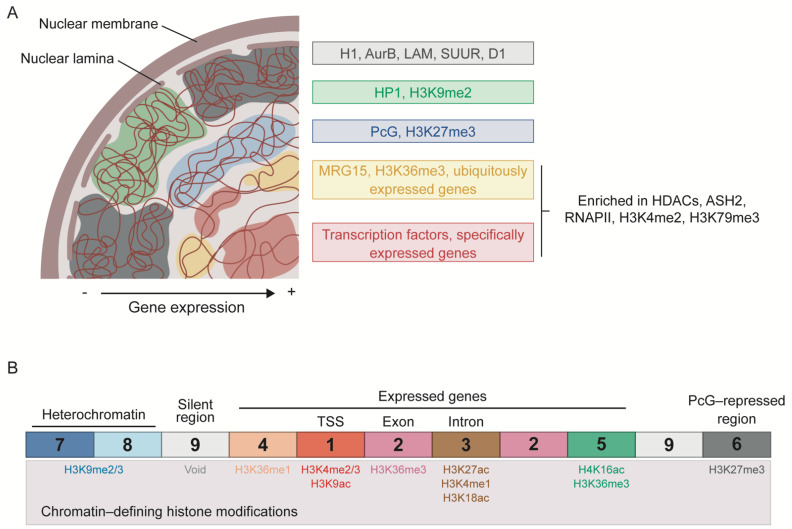
Classification of the chromatin landscape in *Drosophila*. (**A**) Chromatin segmentation into 5 types according to combinatorial protein binding. BLACK, GREEN, and BLUE chromatin types correspond to repressed and silenced domains, whereas YELLOW and RED chromatin types represent active regions. The more repressed regions tend to localize at the periphery of the nucleus, with BLACK and GREEN chromatin interacting with the nuclear lamina. Only the most characteristic components of each chromatin type are indicated. Based on Filion et al. [13]. (**B**) Division of chromatin into 9 states attending to histone modification patterns. States 1, 2, and 5 are associated with actively transcribed genes; states 3 and 4 with putative regulatory regions; state 6 with PcG-repressed regions; states 7 and 8 with heterochromatin; and state 9 corresponds to silent regions. The most enriched histone modifications are indicated for each chromatin state. Based on Karchkenko et al. [92].

**Figure 2 cells-10-02362-f002:**
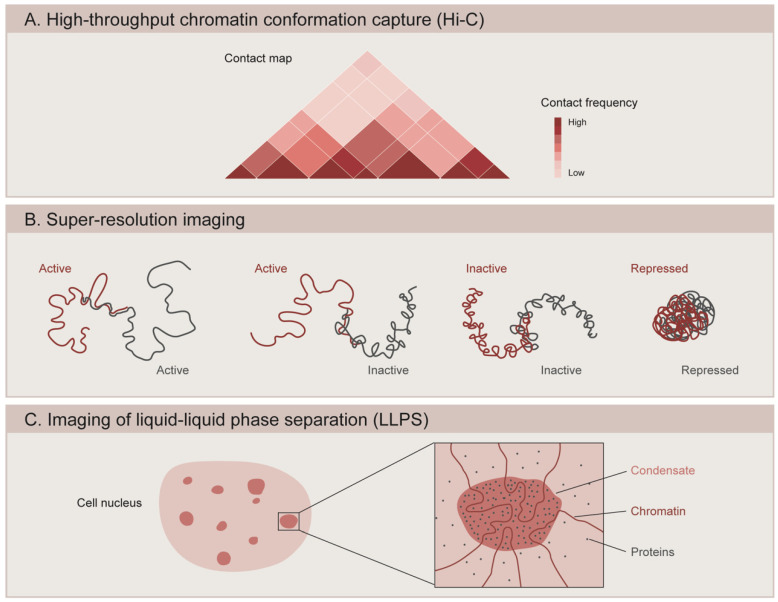
Methods to study the 3D organization of the genome. (**A**) High-throughput chromatin conformation capture (Hi-C) generates contact maps that represent the interaction frequency between genomic loci. Studies using Hi-C have revealed that chromatin is organized into topologically associating domains (TADs). (**B**) Super-resolution microscopy is used to image the spatial organization of different chromatin domains: transcriptionally active (Active), inactive (Inactive), and Polycomb-repressed (Repressed). While active and inactive regions can partially intermix with one another, repressed domains show a more compact configuration and do not overlap with other neighboring domains. (**C**) Microscopy-based methods are used to assess the ability of chromatin components to form condensates through liquid–liquid phase separation (LLPS). Condensate formation is mediated by the biochemical properties of the macromolecules and their interactions.

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
