# Peer review of "Chromatin Organization and Function in Drosophila"

_cells, 2021, doi:10.3390/cells10092362_

Round 1

Reviewer 1 Report

Overall, this is a very comprehensive and well-written review of chromatin organization in the fly system, covering topics from nucleosome structure to nuclear compartmentalization due to liquid-liquid phase separation. I applaud the authors’ ability to provide a balanced perspective on many controversial or open questions by discussing opposing findings. I only have a few suggestions for improvement, as outlined below:

  1. Line 61 “Beyond the binary classification of chromatin into euchromatin 61 and heterochromatin and based on a combinatorial signature of bound proteins, Filion et 62 al. proposed the segmentation of the fly genome into five principal chromatin types [13] 63 (discussed below)”, and Figure 1 and section 5.1 – authors may also want to mention and perhaps briefly discuss that the chromatin landscape has also been sub-dividied into 9 states (Kharchenko PV et al. Nature 2011).

  1. Lines 159-161 “Recent studies have confirmed that NMAD-1 (MT-A70 family) and DMAD (DNA 6mA demethylase, TET ortholog) are 6mA demethylases in C. elegans and D. melanogaster, respectively [69].” - it may be helpful here to expand a bit more on the proposed function of 6mA modification in Drosophila, especially in the context of the general proposed functions of DNA methylation.

  1. 5.2.3. D organization of the Genome” –should probably be “3D”, not “D”.

  1. Line 246 “These five types of chromatin have been related to physical domains of chromosome 246 folding, a clear example being the PcG bodies or PcG-bound domains [89,90].” – this sentence is a bit confusing, and it would help if more detail was provided on how PcG bodies represent chromatin color relating to physical chromosome folding.

  1. For the paragraph within lines 361-387, discussing whether TADs are real physical entities, the authors may be interested in mentioning studies that appears to find good correlation between mapped TADs and banding pattern of polytene chromosomes (Eagen KP et al. Cell 2015, Ulianov SV et al. Genome Res 2016), which provides more evidence that TADs are “real”.

  1. Section 5.4 on nuclear lamina – the authors should consider adding a brief discussion of what’s known on nuclear pore-chromatin interactions since nuclear pores constitute another major protein complex in the nuclear envelope that can interact with chromatin. This topic has been studied by a few labs over the last decade, and just for the fly system, see for instance Kalverda B et al. Cell 2010, Vaquerizas JM et al. PLOS Genetics 2010, Pascual-Garcia P et al. Mol Cell 2017, Ilyin AA et al. NAR 2017, Gozalo A et al. Mol Cell 2017.

Author Response

Please the attachment.

Reviewer 2 Report

This is an interesting and very detailed review focusing on Chromatin organization and function in Drosophila.

Major Points:

However, I have some concern about the conclusion part.

First, a concrete “Conclusions” paragraph is missed as well as a transition link between the descriptive and “Future perspectives” part. I would add a short conclusive paragraph summarizing the state of art just before the “Future perspectives” in order to better link all the paragraphs.

Second, in the “Future perspectives” part (rows 535-536), I would better explain how the latest technical advances including live imaging in combination with CRISPR/Cas9 technology could improve the resolution of chromatin dynamics in whole genomes.

Third, in the paragraph 3.2 “Core Histones and Their Variants” the following references about the role of H2A.v histone variant and polytene chromosome structure regulation are missing:

  • Prozzillo Y, Cuticone S, Ferreri D, Fattorini G, Messina G and Dimitri, P. In Vivo Silencing of Genes Coding for dTip60 Chromatin Remodeling Complex Subunits Affects Polytene Chromosome Organization and Proper Development in Drosophila melanogaster. Int. J. Mol. Sci. 2021, 22, 4525.
  • Messina G, Damia E, Fanti L, Atterrato MT, Celauro E, Mariotti FR, Accardo MC, Walter M, Vernì F, Picchioni D, Moschetti R, Caizzi R, Piacentini L, Cenci G, Giordano E, Dimitri P. Yeti, an essential Drosophila melanogaster gene, encodes a protein required for chromosome organization. J Cell Sci, 2014. 127(Pt 11): p. 2577-88.

Minor Points:

Row 250: “D organization of the Genome” – Some words are missing or incorrect?
